# Added value of assessing medical students' reflective writings in communication skills training: a longitudinal study in four academic centres

Camila Ament Giuliani Franco,[1] Renato Soleiman Franco,[2] Dario Cecilio-Fernandes ![ORCID],[3] Milton Severo,[4] Maria Amélia Ferreira,[5] Marco Antonio de Carvalho-Filho ![ORCID] [6,7]

► Prepublication history and additional materials for this paper is available online. To view these files, please visit the journal online (http://dx.doi.org/10.1136/bmjopen-2020-038898).

For numbered affiliations see end of article.

**Correspondence to**
Dr Marco Antonio de Carvalho-Filho;
m.a.de.carvalho.filho@umcg.nl

## ABSTRACT

**Objectives** This study describes the development and implementation of a model to assess students' communication skills highlighting the use of reflective writing. We aimed to evaluate the usefulness of the students' reflections in the assessment of communication skills.

**Design** Third-year and fourth-year medical students enrolled in an elective course on clinical communication skills development were assessed using different assessment methods.

**Setting and participants** The communication skills course was offered at four universities (three in Brazil and one in Portugal) and included 69 students.

**Outcome measures** The students were assessed by a Multiple-Choice Questionnaire (MCQ), an objective structured clinical examination (OSCE) and reflective writing narratives. The Cronbach's alpha, dimensionality and the person's correlation were applied to evaluate the reliability of the assessment methods and their correlations. Reflective witting was assessed by applying the Reflection Evaluation for Enhanced Competencies Tool Rubric (Reflect Score (RS)) to measure reflections' depth, and the Thematic Score (TS) to map and grade reflections' themes.

**Results** The Cronbach alpha for the MCQ, OSCE global score, TS and RS were, respectively, 0.697, 0.633, 0.784 and 0.850. The interobserver correlation for the TS and RS were, respectively, 0.907 and 0.816. The assessment of reflection using the TS was significantly correlated with the MCQ ($r=0.412$; $p=0.019$), OSCE (0.439; $p=0.012$) and RS (0.410; $p=0.020$). The RS did not correlate with the MCQ and OSCE.

**Conclusions** Assessing reflection through mapping the themes and analysing the depth of reflective writing expands the assessment of communication skills. While the assessment of reflective themes is related to the cognitive and behavioural domains of learning, the reflective depth seems to be a specific competence, not correlated with other assessment methods—possibly a metacognitive domain.

## Strengths and limitations of this study

► This study details the use of medical students' reflective narratives in the assessment of communication skills.

► The assessment of the depth (profundity) and the themes (topics) of medical students' reflective narratives has an additional value compared with the traditional assessment methods used in communication skills training.

► The method used to assess the depth and themes of medical students' reflective narratives showed good reliability.

► The participants were recruited from a convenience sample and further studies are needed to explore the added value of assessing medical students' reflective narratives in a natural context.

skills.[1] Competent doctors must adapt their communication to the specific needs of their patients.[2] In this regard, for medical students to become competent communicators, they must reflect on their experiences with patients aiming for the self-monitoring of their thoughts and behaviours to improve their performance in further interactions with patients.[1 3] Although reflection is an essential component of developing communication,[4] most communication skills training does not include the assessment of students' reflections in their repertoire of assessment tools.[5] Understanding how assessing reflection may support (or not) the development of communication skills in medical students may offer medical educators a new strategy for improving doctor–patient communication.

Medical students must be aware of patients' needs and willing to adapt their patterns of behaviour according these

## INTRODUCTION

Clinical communication is essential for medical students and must extend well beyond the reproduction of behaviours and

needs and context.[6] Although the learning of some basic behavioural rules can indeed be an excellent starting point, such rules governing behaviour may not suffice for guiding students in the process of navigating the complexity of doctor–patient communication. Each patient is unique and has his or her system of beliefs and singular expectations. Doctors must tailor their communication strategies to match each patient needs while respecting his or her personality and social and cultural background.[7 8] Doctors should adapt their communication styles to each patient by addressing the complexity of human interactions, which includes attending with and regulating their own emotions, understanding the context and identifying potential dilemmas. In mastering communication, doctors should reflect before, during and after each clinical encounter to recognise their limitations and identify areas for improvement while planning how to achieve better outcomes.[4] Therefore, we advocate that educational activities that target the development of medical students' communication skills should include the teaching and assessment of reflection.

Within an educational context, reflection is a process[9] whereby individuals critically analyse their cognitive and behavioural responses to a certain experience and develop a deeper understanding of the experience and themselves. The reflection may start even before the experience starts (reflection-for-action), so that students can achieve a broader understanding of a particular task, which helps them to prepare for action. For example, when students anticipate that the task exceeds their level of competence, they may ask for help.[10] The reflection can also occur during the experience (reflection-in-action). This reflection in action refers to the capacity to address just-in-time information by applying the process of analysis and critics during an event, which may lead to real-time adaptation of the performance. After the end of the experience, students can engage in a reflection-on-action process by reviewing and analysing the event and its course to reach a deeper understanding and elaborate new knowledge.[8] Fostering reflection on-action has been one of the starting points for the development of reflective practices in medical education, from first-year undergraduate classes to postgraduate training.[3 11] For instance, in the context of doctor–patient relationship (DPR), the process of reflection on-action has a vital role in building mental models that become available to be applied in future clinical experiences to enhance emotional awareness, emotion expression and empathy.[4 12–14]

Most of the methods for assessing reflection targets reflection on-action processes, mainly by the use of students' reflective writing.[3 11] Reflective writing supports students' self-monitoring, generates self-awareness[15] and promotes a deeper understanding of patients by allowing the inclusion of biopsychosocial perspectives in next consultations.[16 17] Although reflection on-action

has been considered keen in the development of clinical communication,[4 18] its implementation has a low degree of systematisation and minimal attention has been paid to descriptions of the use of reflective writing as an assessment tool in this context.[4]

Reflective writing can be assessed based on the content or depth of reflection. The content of reflection may be evaluated by theme or category-based analysis.[19–21] For example, Karnieli-Miller *et al* used reflective writing to support the teaching of breaking bad news. In the reflective narratives, the authors identified through theme-based analysis all the elements that were part of the clinical protocol used as a reference during the study.[20] However, the study focused on the content of reflection, but not on the depth of reflection. Moreover, the authors did not compare the results of the assessment of the reflection with those obtained through other methods of assessment. Similar to Karnieli-Miller *et al*, Braverman *et al* used a coded framework for the thematic analysis of third-year medical students' reflective writing on challenges in communicating with patients but also did not assess the depth of reflection.[21] Thus, the studies that have sought to determine the role of reflection in teaching communication have targeted its themes, rather than its depth.

The Reflection Evaluation for Enhanced Competencies Tool (REFLECT rubric), proposed by Wald *et al*, highlights the importance of deep reflection in the development of metacognition and effective patient care[22] and has been widely used to evaluate reflection, particularly reflection on-action processes.[23] These authors organised a multidimensional analysis of reflection that assesses five mandatory items: writing spectrum, presence, description of conflict, attending to emotions and meaning making.[22] These five items can be classified using a Likert scale ranging from 0 to 3, according to four different reflection levels (from non-reflective to critically reflective), which correspond to the depth of reflection. This assessment model distinguishes between written texts with only superficial reflection (descriptive) and those with a high density of reflective elements. Although the REFLECT rubric was used successfully in assessment strategies for different learning activities involving reflective writing in medical education, its use in communication skills training must be stimulated and better analysed.[4 11]

Communication training traditionally applies a combination of Multiple-Choice Questionnaires (MCQs) and Objective Structured Clinical Examination (OSCE) stations to assess students' cognitive knowledge and check students' performance.[24 25] Previous research shows a low correlation between the MCQ and OSCE scores, which suggests that, indeed, these methods are assessing different competencies.[26–28] Communication teachers should take advantage of these different scores and provide specific feedback targeting knowledge and/or behaviour. Since cultivating reflection skills is also relevant to the process of becoming a competent communicator, communication trainers should implement

assessment strategies that target reflection skills to create an opportunity to provide feedback on this competency.[29]

There is a lack of research exploring the impact of reflection on the learning of communication skills. The use of the reflective capacity in the teaching and assessment of communication skills, namely, in scenarios related to practice, must be encouraged once it can foster students' professionalism, critical thinking and attitudes.[3 4 12–14 23] Reflective capacity, as a metacognitive process, surpasses (but includes) cognitive and behavioural elements. Understanding the level of correlation between the scores for reflection and the scores for traditional assessments, such as MCQs and OSCEs, potentially contribute to the discussion regarding the role of assessing reflection in communication skills training. Therefore, we raise the following research questions: Is the assessment of reflective writing correlated with cognitive (MCQ) and behavioural (OSCE) assessment methods?

To address these questions, we report the development of a model for assessing the reflection on-action of medical students in the context of communication skills training by applying two methods to evaluate students' reflective writing (themes and depth). We also compare the assessment of reflective writing with other traditional methods (ie, MCQ and OSCE) to understand the added value of assessing the reflection process using these two methods. Understanding the added value of assessing students' reflective writing may contribute to clarify the importance of reflection in the process of honing communication skills to improve doctor–patient communication and support its future application in learning activities.

## METHODS
### Overview

This longitudinal observational study was carried out at three different Brazilian universities (one course at each university in 2015) and one university in Portugal (one course in 2016). Data collection occurred during these elective courses in clinical communication. Each course comprised five modules (25 hours in total) conducted over 2 months. The elective discipline did not disturb students' academic trajectory and occurred in parallel to the regular learning activities. It is worth mentioning that, although this course did not involve practice with patients, all of the students had clinical encounters with patients in hospitals and primary care settings during their regular academic activities.

The Calgary-Cambridge Guide to Communication[30 31] and Patient-Centred Medicine[32] were the conceptual and theoretical models behind this elective communication skills' course. The contents of the first four modules comprised the steps of consultation: (1) initiating the session, (2) gathering information, (3) explanation and planning, (4) closing the session and last one included the (5) breaking bad news. These contents and models were employed as supportive frameworks, and students were not encouraged to follow them as behavioural

protocols. The main focus of the course was on the need to reflect and adapt communication strategies to patients' needs and students' communication style. Each module of the course was structured following four steps: (1) presentation of the content via reflective, small-group discussions, (2) simulation activities with simulated patients; (3) reflective debriefing and (4) summary of the learning points and preparation for next modules.[33] The course did not have a module about theoretical assumptions of reflection or reflective writing, but the instructor of the course structured the discussion of the content (step 1) and debriefing (step 3) using the Gibbs Reflective Circle.[33]

The cases selected for simulation involved clinical scenarios about common health problems with contextual or emotional challenges. For example, in one scenario, an apparently healthy woman asked for a preconception consultation regarding planning for pregnancy. The woman had a history of sexual abuse (between the ages of 11 and 13) by her uncle. She was neglected by her family even after informing her parents about the abuse. This scenario is very emotional and, unfortunately, represents a common occurrence in primary care settings where the students have their clinical training. The learning objective of this scenario is to consider the patient as a whole (one of the main principles of patient-centredness), obtain biopsychosocial information and address emotions (discuss empathy and affective reactions). In preparation to engage with the scenarios, students are stimulated to reflect in action and develop self-awareness and active listening skills, both competencies are among the pillars of one of the theoretical references of the course. During the debriefing of this and other cases, the facilitator stimulated a profound, horizontal and collaborative discussion about the different elements and emotions involved in dealing with the simulated encounter. The facilitator actively invited students to take different perspectives. Every session ended with the elaboration of an action plan aiming to improve student's future performance and provide better patient care. A detailed discussion of the course has been previously published.[34]

### Participants

A convenience sample of third-year and fourth-year medical students at four universities were invited to participate in the study by email. For the sample recruitment, a class representative of the students in the third or fourth-year sent an email to their colleagues inviting them to participate in the course. No financial incentives were given for their participation. A total of 69 participants (20 at University 1—Brazil, 12 at University 2—Brazil, 30 at University 3—Brazil and 7 at University 4—Portugal) agreed to participate. The participants joined a course containing five encounters with a total of 25 hours. The 69 participants were assessed at the end of the course with an MCQ and OSCE on communication skills. The participants were invited (but not obligated) to write a reflective piece, and 37 students produced texts.

## Material: assessment instruments

We compared three different assessment methods: a cognitive test based on an MCQ, an examination of communication skills based on the OSCE, and an assessment of reflection through reflective writing. The MCQ and OSCE were administered after the last meeting of the course on communication skills. The reflective writing was optional and could be undertaken by the students at any point during the communication skills' course. We decided that the reflective writing would be optional to understand the students' disposition to engage with this assessment method.[35]

The MCQ consisted of 63 items about clinical communication. The items were based on clinical situations or conceptual issues that were grounded in the Calgary-Cambridge Guide to Communication,[30 31] Patient-Centred Medicine[32] and Kalamazoo Consensus.[36]

The OSCE included six stations specifically designed to assess communication skills. The OSCE was based on the same references of the MCQ (Calgary-Cambridge Guide to Communication,[30 31] Patient-Centred Medicine[32] and Kalamazoo Consensus[36]). Four of these stations had been tested by the authors in a pilot project[37]). To elaborate the six stations, two medical educators with expertise in OSCE and clinical communication collaborated to develop the stations and checklists. The OSCE targets behavioural domains (communication skills) and affective domains (empathy and compassion) both in the context of doctor–patient interactions. According to the blueprint based on the content of the course, the stations assessed students in scenarios in which they must break bad news to a patient's family, break bad news to a patient, gather information to reach a clinical diagnosis, engage in shared decision making, address moral conflicts and care for a patient with multiple complaints. There was one observer for each OSCE station who was responsible for filling out the assessment checklist. These checklists consisted of between six and 14 items depending on the station. Each item on the checklist was then classified on a Likert scale ranging from 0 to 2 points. The final score of each station was obtained by the mean of its items. The OSCE global score was calculated as a mean considering the six stations.

For the reflective writing component, students could choose any aspect of doctor–patient communication that they considered challenging in their clinical practice. The only advice was that students should find a calm place to write—a place that enables them to focus their attention on their writing with as few distractions as possible. Medical students did not take a course on reflection and reflective writing before this study. The students received the following instruction: 'Suggestion for reflection: (1) describe the situation; (2) point out the dilemmas, doubts and questions raised; (3) point out feelings and observations; (4) analyse the situation from different points of view; (5) make a conclusion and (6) suggest a hypothesis. These steps are only a suggestion; you may conduct the reflection in whichever way that you prefer'.

The writing content was related to communication skills and evaluated (1) through the sum of the themes covered in each one of reflections—the thematic score (TS) and (2) through the REFLECT Rubric—the Reflect Score (RS).[22] In the next paragraphs, we describe how these two scores were calculated.

For establishing the TS, two researchers (CAGF and RSF) started a content analysis individually by reading carefully all the reflective writings made by the students. After reading, CAGF and RSF selected the fragments related to clinical communication[38] and generated a single list with all the fragments from the reflections of all students. Next, CAGF and RSF grouped the fragments in thematic categories independently. After, CAGF and RSF met to reach a consensus on the main themes. After the definition of the main thematic categories, CAGF and RSF read each one of the reflective writings for a second time and decided whether each of the themes were present or not. The two researchers assigned point scores accordingly to the presence of a certain theme ('0' for absent and '1' for present). The final TS corresponded to the sum of all the themes approached by the student. Finally, the agreement between the two researchers was evaluated, and, when there was a difference between the two, a final TS was reached by consensus.

The assessment based on the five mandatory dimensions of the REFLECT Rubric followed the guidelines set by the authors of the rubric. The five mandatory dimensions are: description, presence, identification of a dilemma, emotion and the meaning of the experience. Each one of the dimensions are evaluated considering four levels of reflective capacity scored from 0 to 3 (habitual action or nonreflective=0, thoughtful action or introspection=1, reflection=2 and critical reflection=3). The sum of the scores obtained in each dimension was the total RS. Online supplemental appendix 1 presents a fragment of one reflective writing and the application of the assessment to the five dimensions of the REFLECT rubric (see online supplemental appendix 1).

In summary, the TS refers to 'the subject of reflection—number of themes', the RS refers to 'how the reflection took place or the depth of reflection'.

## Analysis

The quality of the MCQ was assessed by internal consistency, items' responsiveness, face, content and construct validity. The face and content validity of MCQ were developed with the support of the group in the Medical Education Department of the University of Porto, which was responsible for the evaluation of high-stakes examinations of the Faculty of Medicine to guarantee the quality of the items. Three experts in communication (one of them is an external member of the University) assessed and approved the assessment regarding its content. The internal consistency of the items was evaluated by Cronbach's alpha. The responsiveness and construct validity were evaluated according to a published study, in which this MCQ test was applied.[34] The items' responsiveness

was considered adequate once the score before and after a course on communication improved significantly. The mean of improvement was 18.9% (95% CI, ranges from 15.8% to 22.1%) (p<0.001). The MCQ (pre and post-test) was applied to medical students who attained the same communication course at four universities. The improvement in the scores after the course did not show differences among universities (p=0.102). Thus, the results indicate an acceptable construct validity.

The psychometric quality of the OSCE was evaluated by validation of the content (applying the principal component analysis for dimensionality) and internal consistency. Dimensionality was assessed using a scree plot, and the number of components was assessed according to the 'elbow rule'. An element or item was considered to contribute to a principal component when it had a correlation value higher than 0.30. Internal consistency was evaluated using Cronbach's alpha (Cronbach 1951). Acceptable values for internal consistency were considered to be higher than 0.7. The linear associations between the assessment methods were assessed using the Pearson's correlation considering missing complete at random to handle with missed correlations. It was also provided a 95% CI for the Pearson's correlation to present the precision of the correlation.

To measure agreement between researchers, we used the intraclass single average value for absolute agreement. The inter-rater agreement rate was calculated for encoded fragments (TS) and for the RS. NVivo software (V.11.3.2 for Mac) was used for qualitative data analysis, while the SPSS, V.25.0, was used for quantitative data analysis.

Participant consent was requested in the form of an informed consent before the participation in the communication skills course. Signed written consent forms were completed by all participants.

### Patient and public involvement
There is no patient involved in the study.

### RESULTS
Sixty-nine students followed the courses and were included in the study. Fifty-five of the students were women (79.7%), and the mean age of participants was 23.5 years (SD 2.495). Fourth-year students were the largest cohort (69.6%). All participants (69 students) underwent the MCQ and OSCE examinations, and 32 students also performed the reflective writing.

### Quality of the instruments
The MCQ examination had a Cronbach's alpha of 0.697. For the six OSCE stations, the lower Cronbach's alpha level was 0.702, and the higher was 0.815. The Cronbach's alpha of the OSCE global score was 0.633. Considering one component (OSCE global score), the factor loads of the OSCEs stations were higher than 0.3 Table X).

**Table 1** Example of fragments according to thematic categories

| Thematic categories | Fragment example |
| --- | --- |
| Non-verbal | 'I quickly noticed a strange, slightly frightened look on his face…'. |
| Steps of consultation | '…the consultation I performed was… like a questionnaire application…'. |
| Doctor–patient relationship | '…it helps me, mainly to understand how to put the patient's needs and well-being above my own…'. |
| Empathy and respect | 'I believe it is consensual that the attitude of the……is subject to criticism, after all, respect and patience with the patient are prerequisites…'. |
| Humanistic values | '…the way he introduced himself… the attention with which he listened…'. |

The TS had a Cronbach's alpha of 0.784, while the interexaminer correlation for absolute single-measure concordance was 0.907 (two examiners). The RS had a Cronbach's alpha of 0.850 and an interexaminer correlation for absolute single-measure concordance of 0.816 (two examiners).

### Thematic analysis
The thematic categories of the reflections were non-verbal communication, the patient's perspective, steps of communication, DPR, ethics and respect, empathy and altruism and humanistic values (table 1).

### Correlation between instruments
Table 2 shows the correlations between the four different assessment methodologies. There was no correlation between the score for the depth of reflection (RS) and both the MCQ and OSCE scores. The RS was only correlated with the TS. However, the TS score was positively correlated with the MCQ score (0.439; p=0.012) and the OSCE score (0.412; p=0.019).

### DISCUSSION
The assessment of the depth and themes of reflection on-action provides a different perspective on the teaching and learning of communication skills. We found a positive correlation between the content of the students' reflections with their performance on a cognitive test and OSCE assessment, which suggested that the scope of the reflection was related to the students' knowledge. The lack of correlation between the depth of reflection and cognitive and behavioural tests suggests that reflection could be a particular competence domain.

### Importance of including assessment of depth and content when evaluating reflection
The reflection process ranges from elementary cognitive levels (description, identification, knowledge and others)

**Table 2** Pearson correlations between the different methods of assessment

| Assessment methods | OSCE | 95% CI | P value | MCQ | 95% CI | P value | REFLECT Score | 95% CI | P value |
|---|---|---|---|---|---|---|---|---|---|
| MCQ | 0.396 (n=69) | 0.17 to 0.59 | 0.001* | – | | – | – | | – |
| REFLECT Score | 0.250 (n=32) | –0.11 to 0.55 | 0.168 | –0.219 (n=32) | –0.53 to 0.14 | 0.228 | – | | – |
| Thematic Score | 0.412 (n=32) | 0.07 to 0.66 | 0.019* | 0.439 (n=32) | 0.11 to 0.68 | 0.012* | 0.410 (n=32) | 0.07 to 0.66 | 0.020* |

*The p value was considered a sign of statistical significance when it was lower than 0.05.
MCQ, Multiple-Choice Questionnaire; OSCE, objective structured clinical examination; REFLECT, Reflection Evaluation for Enhanced Competencies Tool.

to higher levels of processing, such as analysis, evaluation, synthesis and creation.[38] Using different methods to assess reflection offers an effective strategy to encourage students to engage in reflective activities and enhances the probability of students reaching deeper levels of reflection.[39] Thus, we agree with Hulsman and advocate for the assessment of reflection in terms of its depth and content (themes) to improve communication skills training.[40]

In the assessment of the reflection themes, teachers map the topics students address in their reflections. We observed that the number of themes addressed by students are linked to both knowledge[38] and practical performance.[41–43] Our results suggest that a broader knowledge base and a bigger repertoire of adequate behaviours help students to respond appropriately to different practical situations. The analysis of reflections that are based on its themes can be applied to assist the evaluation of these learning elements.

Interestingly, the reflection depth seems to be a different competence, not necessarily related to the knowledge level or current performance, but possibly related to the values and attitudes of the student regarding a specific topic.[40] It is possible that assessing the depth of reflective writings, even in a particular context (communication skills in our case), enables the evaluation of a specific domain of competence (reflective competence or reflective capacity). Aligned with this hypothesis, Moniz *et al*[44] showed a lack of correlation between the depth of reflection (RS) and OSCE and MCQ scores of undergraduate medical students. However, in Moniz's study, the assessment methods were not targeting the same competence and the absence of standardisation could explain the observed lack of correlation. In our study, we assessed a singular set of competencies (communication skills) and observed the same lack of correlation between reflection depth and other assessment methods. Thus, even after narrowing the context, the lack of association persists.

Learning is a lifelong enterprise and achieving deeper reflection is crucial to the process of becoming an independent and self-regulated learner.[45] The achievement of deeper reflection requires (1) understanding the context; (2) elaborating on the experience; (3) searching for solutions to the problems posed; (4) acknowledging the different subjects involved and (5) taking different perspectives.[46] Thus, when doctors achieve a deep sense of reflection on their practice they move from a state of being knowledge consumers to become active professionals capable of transforming their reality aiming for a practice based on their values and centred on the patient.[47] We believe that deeper reflection goes beyond applying the knowledge to a fixed situation; deeper reflection incorporates the elaboration of new knowledge, balance of different perspectives, anticipation of challenges and planning of future behaviour.[48]

Adding the depth of reflection to teaching and assessment models may allow teachers to capture students' standpoint, their meaning-making processes and their values.[49] We hypothesise that the depth of reflection, particularly concerning communication skills, could be linked to the domain of 'being a doctor' and the formation of professional identity[50 51] by involving elements that extend beyond the context of daily practice to include belief systems and values, which are not commonly assessed in knowledge tests and OSCEs.

### The risks of assessing reflection
The assessment of reflection introduces the risk of limiting the reflective practice For instance, in our study, the observed lack of correlation with cognitive and behavioural assessments may derive from the failing of reflective writing to comprise all of the complexity related to the doctor–patient communication. In practical settings, when caring for a patient, students' reflective practice involves gathering information; being empathetic and compassionate; becoming aware of the clinical, emotional and social context; and identifying conflicts—all crucial elements of addressing patients' needs to guarantee a patient-centred attitude. As a result, reflection is a complex process that involves emotional, cognitive and moral dimensions. Considering this complexity, we must ponder to what extent the writing reflections are capable of capturing all the elements of students' reflective processes. In addition, our grading system may have driven students to focus on some aspects of the

communication process while disregarding other aspects. Grading reflections can pressure students in scoring. The prevalent culture based on targeting high scores may motivate students to 'play the game' and perform tasks and adopt behaviours to fit the expectations of teachers without engaging in transformative learning.[52] Thus, the lack of a correlation between the reflective capacity and knowledge and behaviour and the limits of assessing reflection must be considered. This lack of correlation cannot be extrapolated to the reflective capacity, which is an important limitation of our study. Nevertheless, it is essential to continue investigating the role of reflection as an assessment method for exploring the potential of reflective practice in medical education.[53]

The risk of adopting a reductionistic approach to reflective practices may be avoided by driving the reflective process beyond the achievement of satisfactory grades and performance towards the questioning of taken-for-granted assumptions. These questionings must include the examination of power relations and social and systemic structures. Thus, the reflective capacity should not only address students' knowledge but also foster students' ability to critically analyse what is assumed to be right or wrong.[47 54–57]

### Limitations

This study is one of the first studies to apply multiple methods of assessment, including the evaluation of reflection on-action; however, its limits must be considered. The sample of this study was small and convenient. Our small sample may have influenced both the qualitative analysis and quantitative analysis. It is possible that larger samples could increase the number of categories and subcategories in the thematic analysis. Moreover, the lower number of assessments using the REFLECT rubric[32] restricts the generalisation of the results. The small sample limits the application of more refined statistical methods, for example, adjusting the results for sample characteristics. As it was self-selected, the sample may represent more knowledgeable and motivated students, which may influence both the scores and percentage of students who engaged in the reflective writing (higher than 50%). The fact that the reflection was optional could have attracted students who were naturally reflective, which can also be a confounder. Our results must be confirmed by investigations using non-convenient samples and with a greater number of participants.

The clinical practice involves a complex setting where elements beyond reflective capacity can drive decisions and behaviour, for example, emotional regulation and interpersonal skills. Thus, reflection during an event (reflection in-action) would arguably be more correlated with students' cognitive and behavioural developments. Note that the lack of correlation among the assessment methods relates to reflection on-action and does not relate to reflection in general. To broaden the applicability of reflection as an assessment method, future studies also need to focus on assessing reflection in-action processes.

The assessments were reliable and consistent but limited in terms of reproducibility owing to the number of assessments made. Our method of assessing reflection (reflective writing) could be an element of bias since studies show different results when different reflection methods are used. For example, when reflecting in interviews, students may show levels of reflection that are different from those shown in reflective writing.[58] The current generation of students has a range of preferences when it comes to learning and methods of expression, and many do not have strong skills in written expression.[59] Thus, reflective depth can indeed be associated with students' writing skills.[60] In this way, some authors suggest diversification of reflective registers using alternatives such as digital storytelling. Thus, the use of writing to assume the depth of reflection has an important bias to be considered. Drawing definite conclusions about students' reflectiveness from only one source of reflective material may be biased.

Few studies apply multiple methods to assess communication skills, mainly studies that evaluate reflection. Although the results of this research highlight the assessment of reflections and promote discussions on its use for communication skills training, our assumptions and the limitations of this research may be considered.

### Practical implications

Becoming a good communicator is one of the challenges posed to medical students. Communication training already embraces a body of cognitive knowledge that grounds learning activities. Communication training has also developed different strategies to nurture, check and give feedback on the behaviours and attitudes of medical students during role-playing and simulated or real clinical encounters. However, becoming a good communicator is a life-long process, and, after leaving medical school, junior doctors have to take control of their learning process. Developing a reflective mindset that is capable of evaluating current behaviour—its roots, professional and personal consequences, and emotional impact—will allow junior doctors to transform their understandings and attitudes towards more patient-centred care. Reflection can facilitate this trajectory by supporting medical students during their first steps in becoming autonomous critical thinkers.

### CONCLUSION

This study supports the use of reflective narratives as a complementary assessment method in the context of communication skills training. Assessing the depth of reflection offers a new perspective on students' development and allows the teacher to dive into students' understandings of the value of becoming a good communicator.

**Author affiliations**
[1]Medicine School, Pontifical Catholic University of Parana, Curitiba, Brazil
[2]Medicine School and Post-Graduate Program in Bioethics, Pontifical Catholic University of Paraná, Curitiba, Brazil

³Department of Medical Psychology and Psychiatry, School of Medical Sciences, University of Campinas, Campinas, Brazil
⁴Department of Clinical Epidemiology, Predictive Medicine and Public Health and Public Health and Forensic Sciences, and Medical Education Department, University of Porto Medical School, Porto, Portugal
⁵Public Health and Forensic Sciences, and Medical Education Department, University of Porto Faculty of Medicine, Porto, Portugal
⁶Internal Medicine, University of Minho School of Medicine, Braga, Portugal
⁷CEDAR - Center for Educational Development and Research in Health Sciences, University Medical Center Groningen, Groningen, The Netherlands

**Acknowledgements** We would also like to thank students for their willingness to participate in the course and achieve the assessment methods.

**Contributors** CAGF and RSF participated in the conception, design of the study, acquisition of data, analysis and interpretation of data, writing the final version of the manuscript. DC-F made substantial contributions in the interpretation of data and revising the paper critically for important improvement in whole manuscript. MAdC-F made substantial contributions in the analysis, interpretation of data and in the writing of the final version of the manuscript. MS and MAF made substantial contributions in the conception and design of the study, analysis and interpretation of data. All authors read and approved the final manuscript.

**Funding** Financial support for the authors was provided by scholarships from the Conselho Nacional de Desenvolvimento Científico e Tecnológico (Brazilian National Council of Technological and Scientific Development - 229753/2013-2) and the Coordenação de Aperfeiçoamento de Pessoal de Nível Superior (Coordination for the Improvement of Higher Education Personnel, Brazil -13271/13-0).

**Competing interests** None declared.

**Patient consent for publication** Not required.

**Ethics approval** This research was approved by the Ethics Centre of the São João Hospital Centre of the Faculty of Medicine of the University of Porto (FMUP) and by the Research and Ethics Commission of the Pontifical Catholic University of Paraná (PUCPR).

**Provenance and peer review** Not commissioned; externally peer reviewed.

**Data availability statement** The datasets used during are available from the corresponding author on reasonable request.

**ORCID iDs**
Dario Cecilio-Fernandes http://orcid.org/0000-0002-8746-1680
Marco Antonio de Carvalho-Filho http://orcid.org/0000-0001-7008-4092

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
