## [Reviewer comments · BMJ Open]

ARTICLE DETAILS

TITLE (PROVISIONAL)	The added value of assessing medical students' reflective writings in communication skills training: a longitudinal study in four academic centres.
AUTHORS	Ament Giuliani dos Santos Franco, Camila; Franco, Renato; Cecilio-Fernandes, Dario; Severo, Milton; Ferreira, Maria; de Carvalho-Filho, Marco Antonio

VERSION 1 – REVIEW

REVIEWER	Roger Ruiz Moral Universidad Francisco de Vitoria School of Medicine Madrid Spain
REVIEW RETURNED	05-May-2020

GENERAL COMMENTS	This is an interesting study that expands the assessment of communication skills (CS) through the use of reflection in medical students. For this, the authors offer assessment of the knowledge and practical application of these skills and additionally they explore the reflective students ability through reflective writings, studying the relationships between these different domains. It is a well-founded and highly interesting study to the extent that it helps to understand the importance of incorporating the reflective abilities of students, pointing out a way to apply it in the assessment methodology commonly used in medical education. The work presents an adequate design and use of the methodology, the tools to obtain the different types of information: cognitive (MCQ), behavioral (OSCE) and reflective (thematic and in-depth analysis using the Ward scale) are sufficiently described. The presentation of the results is clear and adapts to the stated objectives. My main concerns in relation to the study, have to do with the way they tried to capture the reflective capacity of the students, which still follows a reductionist approach, and this can affect the results. The authors highlighted this limitation, pointing timidly out other perspectives on the nature of the reflective process, mainly that of Rita Charon (ref 44). Perhaps, for this reason I think the discussion deserves to get into this topic deeper, because this is a key point of the work that could be related to the lack of correlation found by them between the domain of "depth reflection" and the cognitive and behavioral domains. This is something that in my opinion is not sufficiently explained, since human reflection is intuitively influenced by both domains and at the same time it also influences knowledge and behavior. The authors mention the other main limitations of this study, such as the convenience
---

	sample and the problem of written expression of reflection for younger generations. I think the first one could also be related with this lack of correlation (see below) Anyway, in my opinion, this work represents a very valuable effort in recovery for assessment a subject of high interest in medical education in general such as that of the ability to reflect. This is of particular interest in the field of interpersonal relationship assessment and training, for promoting authentic “patient-centered” practice in students and doctors, and not merely the knowledge and use of CS by doctors who are aware of these skills. So I congratulate the authors. Just as suggestions to improve the manuscript: Table 1 with the description of the dimensions of the SR could be avoided since it is found in the original work by Wald et al (ref 13) I miss at least a couple of examples of narrative reflections of different levels (perhaps exposed in Annexes available for consultation) The problem of the small sample size (69), may have limited the richness of the thematic analysis, which in fact seems very simple, showing only 5 very broad categories and no subcategories. Even more problematic are the results of the correlations of the REFLECT score with the other methods, here the sample size is even smaller as it is the convenience sample (32), this and the use of an unobvious statistical test such as the Pearson correlation it can make the results unreliable. I think this also deserves to be discussed further.
--	---

REVIEWER	Lavjay Butani UC Davis, Sacramento, CA, USA
REVIEW RETURNED	30-Jun-2020

GENERAL COMMENTS	Interesting study and one that has an important intended goal- to determine if assessment of students' reflective skills during clinical communications, can add to the overall assessment of their clinical skills. My biggest concern with the study is in the design, which to me is not set up to answer the question. The entire introduction and premise behind the study is that effective communication in the health-care profession is a complex construct and one that involves far more than having and using cognitive skills pertaining to communication. And that one of these non-cognitive skills is being able to 'notice' and 'process' things in the moment so as to make course corrections (reflection in action and reflection for action). This is absolutely true. However, to determine whether assessment of reflective skills can be validated in the context of clinical communication, the authors need to use a validated assessment tool or strategy that measures reflection in action during a communication session. What is being measured in this study is purely 'reflective skills' and not the application of these skills into a potentially emotionally charged clinical context. These 2 are very different from each other. As conceived, the study results are not unexpected at all (measures of cognitive skills don't correlate with reflective skills because these two skills are not
---

	related to each other) and don't answer the question nor are they of practical utility or value. Other less major issues:  1) The validation of the MCQ is not described (if it was validated at all) 2) The OSCE seemingly was only partly validated. Moreover, what constructs was the OSCE designed to measure (knowledge, skills, affective domains etc.). Little detail is provided 3) The context for the reflective prompt is not clear. Were students asked to reflect on a particular challenging communication encounter with a patient? Or during their skills training or course? As written, it appears as a very vague and free-floating instruction. 4) Do students learn about reflection and reflective writing in their educational curriculum? 5) It is unclear if the RS was based on themes specifically related to communication or on the overall submitted narrative. 6) As mentioned earlier, the REFLECT rubric assess reflective skills after the fact (reflection on action) using narratives written after very specific prompts pertaining to challenging/sentinel/disconcerting events. Without more work, it is unclear if this can be applied to answer the investigators' posed question.
--	---

VERSION 1 – AUTHOR RESPONSE

Answers for Reviewer 1, Roger Ruiz Moral

1- This is an interesting study that expands the assessment of communication skills (CS) through the use of reflection in medical students. For this, the authors offer assessment of the knowledge and practical application of these skills and additionally they explore the reflective students ability through reflective writings, studying the relationships between these different domains.

It is a well-founded and highly interesting study to the extent that it helps to understand the importance of incorporating the reflective abilities of students, pointing out a way to apply it in the assessment methodology commonly used in medical education. The work presents an adequate design and use of the methodology, the tools to obtain the different types of information: cognitive (MCQ), behavioral (OSCE) and reflective (thematic and in-depth analysis using the Ward scale) are sufficiently described. The presentation of the results is clear and adapts to the stated objectives.

Authors' response: Thank you for the compliment.

2- My main concerns in relation to the study, have to do with the way they tried to capture the reflective capacity of the students, which still follows a reductionist approach, and this can affect the results. The authors highlighted this limitation, pointing timidly out other perspectives on the nature of the reflective process, mainly that of Rita Charon (ref 44). Perhaps, for this reason I think the discussion deserves to get into this topic deeper, because this is a key point of the work that could be related to the lack of correlation found by them between the domain of "depth reflection" and the cognitive and behavioral domains. This is something that in my opinion is not sufficiently explained, since human reflection is intuitively influenced by both domains and at the same time it also influences knowledge and behavior. The authors mention the other main limitations of this study, such

as the convenience sample and the problem of written expression of reflection for younger generations. I think the first one could also be related with this lack of correlation (see below).

Authors' response: We agree with the reviewers' comment that cognitive and behavioural domains are present in reflection. The assessment method itself and not the reflective capacity could explain the lack of association between the deepen of reflective writing and other methods. The authors reformulated discussion of the risk of assessing reflections as follows:

The assessment of reflection introduces the risk of limiting the reflective practice For instance, in our study, the observed lack of correlation with cognitive and behavioural assessments may derive from the failing of reflective writing to comprise all of the complexity related to the doctor-patient communication. In practical settings, when caring for a patient, students' reflective practice involves gathering information; being empathetic and compassionate; becoming aware of the clinical, emotional, and social context; and identifying conflicts—all crucial elements of addressing patients' needs to guarantee a patient-centred attitude. As a result, reflection is a complex process that involves emotional, cognitive, and moral dimensions. Considering this complexity, we must ponder to what extent the writing reflections are capable of capturing all the elements of students' reflective processes. In addition, our grading system may have driven students to focus on some aspects of the communication process while disregarding other aspects. Grading reflections can pressure students in scoring. The prevalent culture based on targeting high scores may motivate students to “play the game” and perform tasks and adopt behaviours to fit the expectations of teachers without engaging in transformative learning [51]. Thus, the lack of a correlation between the reflective capacity and knowledge and behaviour and the limits of assessing reflection must be considered. This lack of correlation cannot be extrapolated to the reflective capacity, which is an important limitation of our study. Nevertheless, it is essential to continue investigating the role of reflection as an assessment method for exploring the potential of reflective practice in medical education [52].

The risk of adopting a reductionistic approach to reflective practices may be avoided by driving the reflective process beyond the achievement of satisfactory grades and performance towards the questioning of taken-for-granted assumptions. These questionings must include the examination of power relations and social and systemic structures. Thus, the reflective capacity should not only address students' knowledge but also foster students' ability to critically analyse what is assumed to be right or wrong [46,53–56].

3- Anyway, in my opinion, this work represents a very valuable effort in recovery for assessment a subject of high interest in medical education in general such as that of the ability to reflect. This is of particular interest in the field of interpersonal relationship assessment and training, for promoting authentic “patient-centered” practice in students and doctors, and not merely the knowledge and use of CS by doctors who are aware of these skills. So I congratulate the authors.

Authors' response: Thank you for the compliment.

Just as suggestions to improve the manuscript:

3.1 - Table 1 with the description of the dimensions of the SR could be avoided since it is found in the original work by Wald et al (ref 22)

Authors' response: We agreed with the reviewer and removed Table 1.

3.2 - I miss at least a couple of examples of narrative reflections of different levels (perhaps exposed in Annexes available for consultation)

Authors' response: We added in appendix a example of how the reflective writings were assessed.

Appendix 1 - Scoring Reflective Writing using REFLECT rubric:

Reflective Writing: *I felt uncomfortable and it was hard for me stay present in the consultation because of the way the professor informed the diagnosis and managed the patient. Assessing the situation according to what physicians must do, several skills were not fulfilled in the patient care process: attention to patient well-being, autonomy and responsibility to promote better health for patients. The gathering of history by students had no benefit to the patient and only served a didactic function. As the diagnosis is cancer, which is stigmatised and has a very high negative charge (senior physician had performed a prior consultation and obtained all necessary information), it might not be the best time for medical students "to practice" history-taking. After our history-taking, the senior physician discussed the therapy for cancer with students and asked another physician to participate. They discussed the prognosis for the patient, suggested a new protocol in the research phase and assumed results that should not happen. All of these events occurred in front of the patient and their family. Adequate communication is important to adapt communication to each patient. Information must be provided according to subjects' needs and their capacity to understand... "Why to discuss in that way? They discussed uncertain things and affirmed the prognosis and other indications without scientific confirmation. It is difficult to evaluate these complex issues as students due to the scarce theoretical foundation for communication in medical school. The process of assimilation and application of role models prevails if there is no other point of criticism...*

1- Writing Spectrum – Level: Reflection (“movement beyond reporting or descriptive writing to reflecting; i.e., attempting to understand, question, or analyse an event”¹). The fragments disposed of reveal that students wrote beyond the descriptive level. However, they did not explore and criticise the values, believes or assumptions behind the observed behaviour. Thus, this reflection exceeds the descriptive level and achieves reflection but not a critical reflection – the higher level for writing spectrum: *“The gathering of history by students had no benefit for the patient, but only a didactic function.”; “As the diagnosis is cancer, which is stigmatised and has a very high negative charge (senior physician had conducted a prior consultation and obtained all necessary information), it might not be the best time for medical students "to practice" history-taking.”*

2- Presence – Level: Reflection (“sense of writer being largely present”¹) – The students presented the situation including her/himself in the situation, described the situation according to her/his point of view, which enabled an understanding of the participation of the student in the consultation. However, more details are needed to bring the reader to the setting, as expected for the Critical Reflection Level.

3- Description of conflict or disorienting dilemma – Level: Reflection (“description of the disorienting dilemma, conflict, challenge, or issue of concern”¹) – The description includes the disorienting dilemma but does not include a more profound understanding of the “conflict, challenge, or issue of concern that includes multiple perspectives...” as expected for the next level: “Critical Reflection”. There are three main dilemmas: the need to adapt the communication to each patient, the negative role models and the responsibility to patient well-being. All these elements were clearly stated in the text but lacked the necessary detail for Critical Reflection.

4- Attending to Emotions – Level: Thoughtful action (“recognition but no exploration or attention to emotions”¹) – The students described his/her feeling and the narrative transmits his/her difficulty in handling emotions during the situation. However, no exploration was required for the next level of writing (Reflection) and beyond the recognition and insight on emotions necessary in Critical Reflection.

5- Analysis and Meaning Making – Level: Reflection (“some analysis and meaning-making”¹) - The student noticed problems regarding communication and physicians’ attitude. The writing suggests that the students recognised and analysed the situation; however, it could be more comprehensive for achieving Critical Reflection – for example, why did this doctor behave in this manner? The following fragments present some analysis of the student: *“it might not be the best time for medical students “to practice” history-taking...”*; *“To communicate adequately is important to adapt communication to each patient, and the information must be provided according to subjects’ needs and the capacity to understand...”*.

1- REFLECT rubric statements from: Wald, H. S., Borkan, J. M., Taylor, J. S., Anthony, D., & Reis, S. P. (2012). Fostering and Evaluating Reflective Capacity in Medical Education: Developing the REFLECT Rubric for Assessing Reflective Writing. *Academic Medicine*, 87(1), 41–50. The text in Italic correspond to the student reflective writing. The text in bold correspond to the REFLECT rubric items.

3.3 - The authors mention the other main limitations of this study, such as the convenience sample and the problem of written expression of reflection for younger generations. I think the first one could also be related with this lack of correlation (see below)

The problem of the small sample size (69), may have limited the richness of the thematic analysis, which in fact seems very simple, showing only 5 very broad categories and no subcategories. Even more problematic are the results of the correlations of the REFLECT score with the other methods, here the sample size is even smaller as it is the convenience sample (32), this and the use of an unsubtle statistical test such as the Pearson correlation it can make the results unreliable. I think this also deserves to be discussed further.

Authors’ response: We agree with the limitations of our research and clarified them by adding the following sentences. Moreover, to strengthen the results on the correlation, it was provided Confidence Interval (CI) for Person’s correlation to present the precision of the correlation.

In the Limitation:

This study is one of the first studies to apply multiple methods of assessment, including the evaluation of reflection on-action; however, its limits must be considered. The sample of this study was small and convenient. Our small sample may have influenced both the qualitative analysis and quantitative analysis. It is possible that larger samples could increase the number of categories and subcategories in the thematic analysis. Moreover, the lower number of assessments using the REFLECT rubric (32)

restricts the generalisation of the results. The small sample limits the application of more refined statistical methods, for example, adjusting the results for sample characteristics.

In the Table 2:

The Table 2 included the 95% of CI for Pearson's correlation:

Assessment Methods	OSCE	CI 95%	p-Value	MCQ	CI 95%	p-Value	REFLECT Score	CI 95%	p-Value
MCQ	0.396 (n=69)	0.17 to 0.59	0.001*	-		-	-		-
REFLECT Score	0.250 (n=32)	-0.11 to 0.55	0.168	-0.219 (n=32)	-0.53 to 0.14	0.228	-		-
Thematic Score	0.412 (n=32)	0.07 to 0.66	0.019*	0.439 (n=32)	0.11 to 0.68	0.012*	0.410 (n=32)	0.07 to 0.66	0.020*

*the p-value was considered a sign of statistical significance when it was lower than 0.05. CI: Confidence Interval for Pearson's correlation.

Answers for Reviewer 2, Lavjay Butani

Interesting study and one that has an important intended goal- to determine if assessment of students' reflective skills during clinical communications, can add to the overall assessment of their clinical skills.

1- My biggest concern with the study is in the design, which to me is not set up to answer the question. The entire introduction and premise behind the study is that effective communication in the health-care profession is a complex construct and one that involves far more than having and using cognitive skills pertaining to communication. And that one of these non-cognitive skills is being able to 'notice' and 'process' things in the moment so as to make course corrections (reflection in action and reflection for action). This is absolutely true. However, to determine whether assessment of reflective

skills can be validated in the context of clinical communication, the authors need to use a validated assessment tool or strategy that measures reflection in action during a communication session. What is being measured in this study is purely 'reflective skills' and not the application of these skills into a potentially emotionally charged clinical context. These 2 are very different from each other.

Author's response: We agree with the comment of reviewer 2 that the timeframes (before, during and after) for reflection are important and each timeframe has some particularities. The role of reflection in action, on action and for action were not clearly presented. We clarified this part in two aspects: the timeframe for reflection and the impact of the reflective capacity on behavioural skills. In the limitation session, we added that during an action, complex elements, such as emotions, may influence the application of the reflective capacity in practice. Thus, including measures of reflecting in action would strengthen the results and highlight the application of reflective capacity. The following paragraphs were reformulated:

Introduction:

...Doctors should adapt their communication styles to each patient by addressing the complexity of human interactions, which includes attending with and regulating their own emotions, understanding the context, and identifying potential dilemmas. In mastering communication, doctors should reflect before, during and after each clinical encounter to recognise their limitations and identify areas for improvement while planning how to achieve better outcomes [4]. Therefore, we advocate that educational activities that target the development of medical students' communication skills should include the teaching and assessment of reflection.

Within an educational context, reflection is a process [9] whereby individuals critically analyse their cognitive and behavioural responses to a certain experience and develop a deeper understanding of the experience and themselves. The reflection may start even before the experience starts (reflection-for-action), so that students can achieve a broader understanding of a particular task, which helps them to prepare for action. For example, when students anticipate that the task exceeds their level of competence, they may ask for help [10]. The reflection can also occur during the experience (reflection-in-action). This reflection in action refers to the capacity to address just-in-time information by applying the process of analysis and critics during an event, which may lead to real-time adaptation of the performance. After the end of the experience, students can engage in a reflection-on-action process by reviewing and analysing the event and its course to reach a deeper understanding and elaborate new knowledge [8]. Fostering reflection on-action has been one of the starting points for the development of reflective practices in medical education, from first-year undergraduate classes to post-graduate training [3,11]. For instance, in the context of doctor-patient relationship, the process of reflection on-action has a vital role in building mental models that become available to be applied in future clinical experiences to enhance emotional awareness, emotion expression, and empathy [4,12–14].

Most of the methods for assessing reflection targets reflection on-action processes, mainly by the use of students' reflective writing [3,11]. Reflective writing supports students' self-monitoring, generates self-awareness [15] and promotes a deeper understanding of patients by allowing the inclusion of biopsychosocial perspectives in next consultations [16,17]. Although reflection on-action has been considered keen in the development of clinical communication [4,18], its implementation has a low degree of systematisation and minimal attention has been paid to descriptions of the use of reflective writing as an assessment tool in this context [4].

Limitation:

The clinical practice involves a complex setting where elements beyond reflective capacity can drive decisions and behaviour, for example, emotional regulation and interpersonal skills. Thus, reflection during an event (reflection in-action) would arguably be more correlated with students' cognitive and behavioural developments. Note that the lack of correlation among the assessment methods relates to reflection on-action and does not relate to reflection in general. To broaden the applicability of reflection as an assessment method, future studies also need to focus on assessing reflection in-action processes.

2- As conceived, the study results are not unexpected at all (measures of cognitive skills don't correlate with reflective skills because these two skills are not related to each other) and don't answer the question nor are they of practical utility or value.

Author's response: We clarify and reformulated the research question as suggested by the reviewer. The reviewer's opinion, which is related to the lack of correlation between cognitive skills and reflective skills, is supported by references in medical education. However, we suggest that our research could contribute to the discussion of a multiple model of assessment for communication skills. Few researchers discuss the use of reflection, particularly reflective capacity. We also included a discussion on the importance of emotions; for example, when students apply reflection in clinical practice. Reflection, as a metacognitive process, includes the cognitive domain; however, the lack of association could reveal that other elements of reflection as emotions and behaviours may surpasses the cognitive domain of learning as stated by reviewer 2.

The importance of reflective assessment and the research question were better described in the introduction and discussion. The following paragraphs were reformulated:

Introduction:

In this regard, for medical students to become competent communicators, they must reflect on their experiences with patients aiming for the self-monitoring of their thoughts and behaviours and preparing them for further interactions with patients [1,3]. Although reflection is an essential component of developing communication [4], most communication skills training does not include the assessment of students' reflections in their repertoire of assessment tools [5]...

In this regard, for medical students to become competent communicators, they must reflect on their experiences with patients aiming for the self-monitoring of their thoughts and behaviours to improve their performance in further interactions with patients [1,3]. Although reflection is an essential component of developing communication [4], most communication skills training does not include the assessment of students' reflections in their repertoire of assessment tools [5]...

There is a lack of research exploring the impact of reflection on the learning of communication skills. The use of the reflective capacity in the teaching and assessment of communication skills, namely, in scenarios related to practice, must be encouraged once it can foster students' professionalism, critical thinking and attitudes [3,4,12–14,23]. Reflective capacity, as a metacognitive process, surpasses (but includes) cognitive and behavioural elements. Understanding the level of correlation between the scores for reflection and the scores for traditional assessments, such as MCQs and OSCEs, potentially contribute to the discussion regarding the role of assessing reflection in communication skills training. Therefore, we raise the following research questions: Is the assessment of reflective writing correlated with cognitive (MCQ) and behavioural (OSCE) assessment methods?

To address these questions, we report the development of a model for assessing the reflection on-action of medical students in the context of communication skills training by applying two methods to evaluate students' reflective writing (themes and depth). We also compare the assessment of reflective writing with other traditional methods (i.e., MCQ and OSCE) to understand the added value of assessing the reflection process using these two methods. Understanding the added value of assessing students' reflective writing may contribute to clarify the importance of reflection in the process of honing communication skills to improve doctor-patient communication and support its future application in learning activities.

3-Other less major issues:

3.1 - The validation of the MCQ is not described (if it was validated at all)

Author's response: We provided more information about the validity of the MCQ:

In the subheading - Assessment Instruments it was added:

The MCQ consisted of 63 items about clinical communication. The items were based on clinical situations or conceptual issues that were grounded in the Calgary-Cambridge Guide to Communication [30,31], Patient-Centred Medicine [32], and Kalamazoo Consensus [36].

In the subheading - Analysis it was added:

The quality of the MCQ was assessed by internal consistency, items' responsiveness, face, content, and construct validity. The face and content validity of MCQ were developed with the support of the group in the Medical Education Department of the University of Porto, which was responsible for the evaluation of high-stakes exams of the Faculty of Medicine to guarantee the quality of the items. Three experts in communication (one of them is an external member of the University) assessed and approved the assessment regarding its content. The internal consistency of the items was evaluated by Cronbach's alpha. The responsiveness and construct validity were evaluated according to a published study, in which this MCQ test was applied [34]. The items' responsiveness was considered adequate once the score before and after a course on communication improved significantly. The

mean of improvement was 18.9% (confidence interval of 95%, ranges from 15.8 to 22.1%) ($p < 0.001$). The MCQ (pre and post-test) was applied to medical students who attained the same communication course at 4 universities. The improvement in the scores after the course did not show differences among universities ($p = 0.102$). Thus, the results indicate an acceptable construct validity.

3.2 - The OSCE seemingly was only partly validated. Moreover, what constructs was the OSCE designed to measure (knowledge, skills, affective domains etc.). Little detail is provided

Author's response: We included more information about the design and quality of OSCEs in the following sentences:

In the subheading - Assessment Instruments it was added:

The OSCE included six stations specifically designed to assess communication skills. The OSCE was based on the same references of the MCQ (Calgary-Cambridge Guide to Communication [30,31], Patient-Centred Medicine [32] and Kalamazoo Consensus [36]). Four of these stations had been tested by the authors in a pilot project [37]). To elaborate the six stations, two medical educators with expertise in OSCE and clinical communication collaborated to develop the stations and checklists. The OSCE targets behavioural domains (communication skills) and affective domains (empathy and compassion) both in the context of doctor-patient interactions. According to the blueprint based on the content of the course, the stations assessed students in scenarios in which they must break bad news to a patient's family, break bad news to a patient, gather information to reach a clinical diagnosis, engage in shared decision-making, address moral conflicts, and care for a patient with multiple complaints.

In the subheading - Analysis it was added:

The psychometric quality of the OSCE was evaluated by validation of the content (applying the principal component analysis for dimensionality) and internal consistency.

3.3 - The context for the reflective prompt is not clear. Were students asked to reflect on a particular challenging communication encounter with a patient? Or during their skills training or course? As written, it appears as a very vague and free-floating instruction.

Author's response: The context for reflection was better described. We added in the heading Material: Assessment Instruments:

For the reflective writing component, students could choose any aspect of doctor-patient communication that they considered challenging in their clinical practice. The only advice was that

students should find a calm place to write – a place that enables them to focus their attention on their writing with as few distractions as possible.

In the heading Overview it was added:

The elective discipline did not disturb students' academic trajectory and occurred in parallel to the regular learning activities. It is worth mentioning that, although this course did not involve practice with patients, all of the students had clinical encounters with patients in hospitals and primary care settings during their regular academic activities.

3.4 - Do students learn about reflection and reflective writing in their educational curriculum?

Author's response: The following sentence was added:

Medical students did not take a course on reflection and reflective writing before this study.

3.5 - It is unclear if the RS was based on themes specifically related to communication or on the overall submitted narrative.

Author's response: The RS and TS were based on a reflection on communication. The following sentence was added:

The writing content was related to communication skills and evaluated (1) through the sum of the themes covered in each one of reflections – the thematic score (TS), and (2) through the REFLECT Rubric – the reflect score (RS) [22].

3.6 - As mentioned earlier, the REFLECT rubric assess reflective skills after the fact (reflection on action) using narratives written after very specific prompts pertaining to challenging/sentinel/disconcerting events. Without more work, it is unclear if this can be applied to answer the investigators' posed question.

Author's response: The research questions were clarified according to the remoulding of the last paragraph of the introduction.

There is a lack of research exploring the impact of reflection on the learning of communication skills. The use of the reflective capacity in the teaching and assessment of communication skills, namely, in scenarios related to practice, must be encouraged once it can foster students' professionalism, critical thinking and attitudes [3,4,12–13,23]. Reflective capacity, as a metacognitive process, surpasses (but includes) cognitive and behavioural elements. Understanding the level of correlation between the scores for reflection and the scores for traditional assessments, such as MCQs and OSCEs, potentially contribute to the discussion regarding the role of assessing reflection in communication skills training. Therefore, we raise the following research questions: Is the assessment of reflective writing correlated with cognitive (MCQ) and behavioural (OSCE) assessment methods?

Author's response: Moreover, the role of reflection-on-action in skills and behaviours was detailed in introduction to justify the use of reflective writing in the context of teaching communication skills. We aimed to compare different models of assessment (MCQ and OSCE) with reflection on-action (evaluated in two different ways). Depending how reflections are assessed it seems that it can or cannot have correlation with MCQ and OSCE. The limits of using reflection on-action, namely, reflective writing to infer reflective capacity, was clearer stated in the subheading "The Risks of Assessing Reflection". The whole subheading was rewritten to a deeper understanding of this topic:

The assessment of reflection introduces the risk of limiting the reflective practice For instance, in our study, the observed lack of correlation with cognitive and behavioural assessments may derive from the failing of reflective writing to comprise all of the complexity related to the doctor-patient communication. In practical settings, when caring for a patient, students' reflective practice involves gathering information; being empathetic and compassionate; becoming aware of the clinical, emotional, and social context; and identifying conflicts—all crucial elements of addressing patients' needs to guarantee a patient-centred attitude. As a result, reflection is a complex process that involves emotional, cognitive, and moral dimensions. Considering this complexity, we must ponder to what extent the writing reflections are capable of capturing all the elements of students' reflective processes. In addition, our grading system may have driven students to focus on some aspects of the communication process while disregarding other aspects. Grading reflections can pressure students in scoring. The prevalent culture based on targeting high scores may motivate students to "play the game" and perform tasks and adopt behaviours to fit the expectations of teachers without engaging in transformative learning [51]. Thus, the lack of a correlation between the reflective capacity and knowledge and behaviour and the limits of assessing reflection must be considered. This lack of correlation cannot be extrapolated to the reflective capacity, which is an important limitation of our study. Nevertheless, it is essential to continue investigating the role of reflection as an assessment method for exploring the potential of reflective practice in medical education [52].

The risk of adopting a reductionistic approach to reflective practices may be avoided by driving the reflective process beyond the achievement of satisfactory grades and performance towards the questioning of taken-for-granted assumptions. These questionings must include the examination of power relations and social and systemic structures. Thus, the reflective capacity should not only address students' knowledge but also foster students' ability to critically analyse what is assumed to be right or wrong [46,53–56].

The following text was added to the Limitations section:

This study is one of the first studies to apply multiple methods of assessment, including the evaluation of reflection on-action; however, its limits must be considered.

VERSION 2 – REVIEW

REVIEWER	Lavjay Butani University of California Davis, USA
REVIEW RETURNED	15-Aug-2020
GENERAL COMMENTS	All of my questions and concerns have been addressed (the ones that could be)